# Ammonium Lignosulfonate Adhesives for Particleboards with pMDI and Furfuryl Alcohol as Crosslinkers

**DOI:** 10.3390/polym11101633

**Published:** 2019-10-10

**Authors:** Venla Hemmilä, Stergios Adamopoulos, Reza Hosseinpourpia, Sheikh Ali Ahmed

**Affiliations:** Department of Forestry and Wood Technology, Linnaeus University, Lückligs plats 1, 351 95 Växjö, Sweden; venla.hemmila@lnu.se (V.H.); reza.hosseinpourpia@lnu.se (R.H.); sheikh.ahmed@lnu.se (S.A.A.)

**Keywords:** biorefinery lignin, wood panels, sustainable adhesives, adhesive penetration, particleboard properties, formaldehyde emissions

## Abstract

Tightening formaldehyde emission limits and the need for more sustainable materials have boosted research towards alternatives to urea-formaldehyde adhesives for wood-based panels. Lignin residues from biorefineries consist of a growing raw material source but lack reactivity. Two crosslinkers were tested for ammonium lignosulfonate (ALS)—bio-based furfuryl alcohol (FOH) and synthetic polymeric 4,4′-diphenylmethane diisocyanate (pMDI). The addition of mimosa tannin to ALS before crosslinking was also evaluated. The derived ALS adhesives were used for gluing 2-layered veneer samples and particleboards. Differential Scanning Calorimetry showed a reduction of curing temperature and heat for the samples with crosslinkers. Light microscopy showed that the FOH crosslinked samples had thicker bondlines and higher penetration, which occurred mainly through vessels. Tensile shear strength values of 2-layered veneer samples glued with crosslinked ALS adhesives were at the same level as the melamine reinforced urea-formaldehyde (UmF) reference. For particleboards, the FOH crosslinked samples showed a significant decrease in mechanical properties (internal bond (IB), modulus of elasticity (MOE), modulus of rupture (MOR)) and thickness swelling. For pMDI crosslinked samples, these properties increased compared to the UmF. Although the FOH crosslinked ALS samples can be classified as non-added-formaldehyde adhesives, their emissions were higher than what can be expected to be sourced from the particles.

## 1. Introduction

Wood-based panel industry uses almost solely petrol-based adhesives, such as urea-formaldehyde and melamine-urea-formaldehyde. There are two main drivers to replace these synthetic systems with formaldehyde-free bio-based alternatives; lowered formaldehyde limits and the need for sustainability. Formaldehyde has been classified as a human cancer-causing carcinogen (Group 1) since 2004 [1]. The formaldehyde limits have steadily been lowered since, the latest example being the Federal Ministry for the Environment, Nature Conservation and Nuclear Safety of Germany changing their legislation, and effectively lowering the European E1 emission level of 0.10 ppm to 0.05 ppm (EN 717–1) in Germany [2]. These low limits can be reached by modifying the existing systems, as an example by adding formaldehyde scavengers, such as urea, or by using natural compounds, such as waste wood bark floors [3]. Active powdered bark has been found to be able to reduce the free formaldehyde release to about 75% on plywood [4]. Bio-based adhesives can be fully formaldehyde-free and have the additional benefit of increased sustainability. However, to reach 100% bio-based formulations is challenging, and the focus has been on increasing the content of bio-based material with a stepwise approach. The properties and reaction time are enhanced by using synthetic, reactive crosslinkers, such as polyethyleneimines (PEI) for proteins [5], aldehydes for lignins and tannins [6], and the universal crosslinker, polymeric diphenylmethane diisocyanate (pMDI) for most bio-based materials [7,8].

The most researched biomaterials for wood panel adhesives are proteins (e.g., soy and wheat), starches, tannins, and lignins [9,10,11,12,13]. Out of these, lignin is the one with high volumes and only a few value-added applications. Out of the 50 million tons of lignin produced worldwide annually, the majority is used as a combustion fuel in pulp mills, and only 2 wt % is used commercially [14,15]. Lignin consists of randomly crosslinked phenylpropanoid units linked to each other by C-O-C and C-C bonds. The word “lignin” refers both to natural lignin found in plants, as well as to the by-product from pulping, papermaking, and biorefinery processes that generate energy, fuels, and materials [16]. The increasing amounts of lignin available from biorefinery processes make it important to find suitable value-adding applications [17]. Lignins from sulfite processes, lignosulfonates, are water-soluble almost over the entire pH range [18]. In the sulfite pulping process, aqueous sulfur dioxide is combined with counter ions, such as ammonium (NH^4+^) or sodium (Na^+^). Sulfonic acid is introduced to the α-carbon atoms of lignin, leading to hydrolysis, which increases the hydrophilicity, and thus the solubility of lignin in the pulping process. This is due to the cleavage of the linkages between phenylpropane units that frees phenolic hydroxyl groups, which are later available for further reactions [19].

One of the main uses of lignin residues is lignin-phenol-formaldehyde (LPF) resins, where lignin is used to partially replace phenol. The phenol replacement amounts are typically below 50%, as the addition of lignin lowers the reactivity of the resin, leading to increased reaction times [20]. Lignin modifications, such as methylolation and phenolation, can be used to increase lignin reactivity with industrially acceptable cost [11,21]. However, this is not enough, especially if unpurified sulfur-containing technical lignins (Kraft lignins and lignosulfonates) are used. In order to use these lignins as main components in adhesive formulations, the low reactivity needs to be compensated by using a suitable crosslinker, especially for particleboard and fibreboard production, where the production speed is a critical parameter [11].

Furfuryl alcohol (furan-2-ylmethanol) is a bio-sourced alcoholic heterocyclic compound from the furan family and can be produced by the reduction of furfural. In the presence of a catalyst at elevated temperatures, furfuryl alcohol polymerizes into a dark-colored, insoluble polymer through a complex polycondensation process that is influenced by many factors, such as the strength and type of the acid catalyst, temperature, and presence of water [22,23,24]. Furfuryl alcohol requires very acid conditions for curing, which is problematic for wood-based materials as the low pH and high residual acidity hydrolyzes the holocellulose of wood at the interface. However, there are some studies suggesting that it might also be possible to have reactive furfuryl alcohol at alkaline conditions [25]. When blended in plasticized form into furfuryl alcohol during its polymerization, lignin is capable of creating linkages to furfuryl alcohol to form thermoset resins. In a study by Guigo et al. [26], two addition rates of straw and grass lignin from the soda pulping process were tested, 20 and 30 wt %. Zhang et al. found that furfuryl alcohol could be reacted with glyoxal before reacting with lignin. The addition of epoxy resin (3, 6, and 9%) further improved the performance of the lignin-glyoxal-furfuryl alcohol particleboard adhesives [27]. Lignins are also proposed to react with furfuryl alcohol precursor, furfural, at high acidic conditions, where lignin and furfural are proposed to replace phenol and formaldehyde in PF resin formulations, respectively [28].

Polymeric 4,4′-diphenylmethane diisocyanate (pMDI) is commonly used as a sole adhesive in the wood panel industry, especially in the production of oriented strand board (OSB) and medium-density fibreboard (MDF) [29]. For particleboards, it is the only industrial formaldehyde-free adhesive used in large industrial scale [30], standing approximately 1% of the annual European panel production [31]. However, pMDI is very expensive and needs to be used in stabilized form due to its high reactivity towards moisture [9]. It can be used as a crosslinker for bio-based materials, such as proteins [32] and mixtures of tannins and lignins [33,34]. As pMDI is a strong adhesive on its own, the final test results of these combinations can be more dependent on the pMDI amount, and the contribution of the bio-based material can remain unclear [30]. The previous study on the performance of one-layer particleboard bonded with glyoxalated lignin combined with pMDI showed superior internal bond (IB) strength in dry and boiled conditions at a mixture level of 60/40 wt % glyoxalated lignin/pMDI [7]. Lei et al. (2008) reported that the internal bond strength of particleboards produced with glyoxalated lignin, mimosa tannin, and pMDI, up to 80 wt % natural polymers, met the requirements for interior grade panels.

The utilization of proper crosslinker is an important factor for developing a lignin-based adhesive that can meet the reaction speed and adhesion strength requirements of the wood panel industry. High molecular weight ammonium lignosulfonates (ALS) have high absorptivity, making them suitable to be used as dispersants. The high amount of hydroxyl group makes ALS suitable for adhesive applications, such as reaction with isocyanates to form urethanes [35]. Thus, ALS has the potential to be used as an adhesive with additional dispersive properties to enhance the spreading of the other components. Therefore, this work was conducted with the aim to study ALS adhesives with synthetic (pMDI) and bio-based (furfuryl alcohol) crosslinkers for particleboard making. The effect of tannin addition into the adhesive mixture was also studied for both crosslinkers. The adhesives were initially evaluated by means of differential scanning calorimetry (DSC) and veneer gluing to get details on their curing behavior, interaction with wood tissues (penetration), and shear bond strength. Three-layered particleboards were then tested in terms of their physical, mechanical, and formaldehyde emission properties, and the results were compared with those bonded with a commercial adhesive.

## 2. Materials and Methods

### 2.1. Chemicals and Adhesive Formulations

Commercial ammonium lignosulfonate (ALS) and mimosa tannin (m) were kindly provided by Borregaard (Neuss, Germany) and Silvachimica S.r.l. (San Michele Mondovì, Italy), respectively. ALS was derived from fermented Norway spruce (*Picea abies*) wood sulfite liquor. Sulfuric acid (H_2_SO_4_, 98%) and furfuryl alcohol (FOH, 98%) were purchased from Sigma Aldrich (Saint Louis, MO, USA). Melamine reinforced urea-formaldehyde (UmF) with a solid content of 68% was purchased from AkzoNobel (Stockholm, Sweden). Ammonium sulfite, with a concentration of 30%, was purchased from Yara (Köping, Sweden). Polymeric 4,4′-diphenylmethane diisocyanate (pMDI) adhesive (I-Bond PBEM 4352) was purchased from Huntsman (Everberg, Belgium).

Table 1 shows details on the various adhesive formulations for testing of 2-layered veneer samples and particleboards based on ALS and mimosa tannin (m) with FOH and pMDI as crosslinkers. For 2-layered veneer testing, the same application amount (100 g solids/m^2^) was used. For particleboard tests, a base application amount of 12% (wt % on dry particles) was used for the UmF and ALS references. For the formulations with crosslinkers, half of the base (6% of ALS or ALS + m) was replaced with the crosslinker. As a typical application amount of these crosslinkers is around 4%, half of this, i.e., 2 wt % on dry fibers (25% of the total adhesive amount), was used as the crosslinker amount, leading to a final application amount of 8%.

For all formulations, the dry ALS powder, with or without tannin (m) powder, was first dissolved in water to form the base solution. Lignin (or lignin and tannin) to water ratio was 1:2, and the final solid content for all formulations with crosslinkers was 48% ± 2%. For the formulations with the bio-based crosslinker FOH (ALS-FOH and ALS-FOH + m), the base solution consisting of ALS or ALS + m was first heated to 70 ± 3 °C under constant magnetic stirring, after which the FOH was added. The pH was adjusted to 2 with 30% H_2_SO_4_ (2% *w/w* of solids), and the reaction proceeded for 45 min for adhesive evaluations on veneer samples. An identical procedure was applied for the preparation of the same adhesives for particleboard manufacturing. A shorter reaction time (30 min) was required to keep the viscosity of the adhesive at a spray-able range (<300 mPa·s) for the lab-scale semi-manual blender used in this study. Viscosity at the end of the reaction was controlled, using a TQC rotational portable viscometer (DV1400, Capelle aan den IJssel, The Netherlands). Spindle 2 was used for values below 400 and spindle 7 for values around 2000. For the adhesives with the synthetic crosslinker ALS-pMDI and ALS-pMDI + m, pMDI was added to the ALS-water solution (concentration 50%) just before the application of the adhesive for veneer gluing. It was possible to apply these adhesives containing pMDI within 25 min. For particleboard production, the base solutions (ALS and ALS + m) were sprayed first on the wood particles, and then pMDI was sprayed separately to avoid a preliminary reaction. As references for the various tests (2-layered veneer samples, particleboards), industrial UmF resin, with a melamine content of 5%, and 50% ALS-water solution alone was used at 12% (based on the dry weight of wood). For UmF, 3% (*w/w* of dry resin) of ammonium sulfite with a solid content of 45% was added as a hardener.

### 2.2. Differential Scanning Calorimetry (DSC)

Curing behaviors of the adhesive formulations (except for UmF) were determined using a NETZSCH STA 409PC (Bayern, Germany) instrument. A total of 5–10 mg of freshly blended mixtures was heated from 25 to 225 °C at a heating rate of 20 °C min^−1^ under a nitrogen flow of 10 mL/min, as explained previously [36].

### 2.3. Shear Bond Strength

The shear bond strength of all the adhesive formulations shown in Table 1 was evaluated by tensile shear strength using a lap joint test according to EN 205 [37] with Instron universal testing machine (Buckinghamshire, England). Beech veneers with a moisture content of 12 ± 1% were used to produce 2-layered samples of dimensions 100 × 20 × 3.2 (L × W × H) mm^3^. Adhesives’ application amount was 100 g of solids per m^2^ to an area of 20 × 20 mm^2^. The samples were pressed in a single daylight press at a temperature of 150 °C for 90 s and conditioned before testing. The loading speed (constant) was set to 2 mm/min, and a total of ten repetitions of each adhesive formulation were tested.

### 2.4. Fourier Transform Infrared (FTIR) Spectroscopy

The chemical structure of the adhesive formulations crosslinked with pMDI was analyzed with a P-Elmer FTIR Spectrometer (Seer Green, United Kingdom). The evaluation was performed between 4000 and 800 cm^−1^ at room temperature, accumulating 10 scans with a resolution of 4 cm^−1^, as described previously [38]. The spectra were collected from bondlines of tensile shear strength test samples in order to confirm the incorporation of ALS in the cured adhesives by chemical reaction with pMDI, i.e., the formation of urethane linkages.

### 2.5. Adhesive Penetration

To observe the bondline thickness and adhesive penetration into the porous wood structure, 30–40 µm thick sections exposing a bondline with a cross-sectional surface at random longitudinal positions of the 2-layered veneer samples were cut using a sliding microtome (WSL, Birmensdorf, Switzerland). The sections were placed unstained on glass slides and viewed under a motorized BX63F light microscope equipped with the DP73 color CCD cooled camera (max. 17.28 megapixel) and the software program cellSens DIMENSION 1.18 (all Olympus, Lund, Sweden). For each adhesive formulation, a minimum of 10 microtome slices were cut in order to obtain microscopy pictures. Random regions of interest with dimensions of 1100 × 600 (W × H) µm^2^ were selected to measure the bondline thickness and the maximum penetration (MP), as explained in previous studies [39,40]. A total number of 30 MP measurements were performed. The used software was able to distinguish between the darker adhesive parts from the lighter colored wood tissues, and hence to calculate the desired statistical parameters. The average of 5 deepest detected adhesive objects in the interphase region was selected to measure the MP using the equation:
$$MP=\frac{\sum_{i}^{5}\left(y_{i}+r_{i}-y_{0}\right)}{5}$$ where MP is the maximum depth of penetration (µm), *y_i_* is the centroid of adhesive object *i* (µm), *r_i_* is the mean radius of adhesive object *i* (µm), and *y*_0_ is the reference y-coordinate of the bondline interface (µm).

In addition, for each of the 30 selected regions, a bondline thickness was measured as the thickness of the region around the center of the bondline, where the adhesive was continuously present.

### 2.6. Particleboard Manufacturing and Testing

Standard industry core and surface layer wood chips from softwood species (Scots pine and Norway spruce) were oven-dried to 1.0 ± 0.3% moisture content. Dried chips were then blended with the adhesive formulations, shown in Table 1, in a drum fitted with a gun triggered by an air compressor. No wax or any other additives were applied for the manufacturing of the exploratory laboratory boards. The resinated chips were placed in a mold with dimensions of 500 × 450 mm^2^ to produce 3-layer particleboards (65% core share) with a thickness of 12 mm and a target density of 620 kg/m^3^. All boards were pressed at a temperature of 190 °C for 270 s using 12 mm metal stops in a single daylight hot press (Ake, Mariannelund, Sweden). After cooling to room temperature, the boards were cut to 400 × 400 mm^2^ dimensions and conditioned at 65 ± 3% relative humidity and 20 ± 2 °C before testing. Two particleboard panels were used to evaluate the physical and mechanical properties for each adhesive system. For internal bond (IB) measurements, eight squares with a side length of 50 mm were cut and tested according to EN 319 [41]. For moduli of elasticity (MOE) and rupture (MOR), 6 pieces with a thickness of 50 mm and length of 210 mm were cut and tested according to EN 310 [42]. Mechanical tests were performed with an Instron universal testing machine (Buckinghamshire, England). Eight squares with a side length of 50 mm were cut and immersed in water with a temperature of 20 °C for 2 h and 24 h, and the respective thickness swelling values were calculated according to EN 317 [43].

### 2.7. Formaldehyde Emission Testing

Formaldehyde emission measurements with 0.044 m^3^ Dynamic Micro Chamber (DMC) were performed according to ASTM D 6007–14 [44] standard method (10.39 m^2^/m^3^ of particleboard at 25 °C, 50% relative humidity, and an air exchange rate of 12.19 h^−1^), as described previously [45]. The ASTM D 6007–14 method was chosen for being fast and for having a good correlation to EN 717–1 chamber method [46].

### 2.8. Statistical Analysis

Mean values and standard deviations were calculated with Microsoft Excel. Standard deviations (sd) were calculated using the formula:
$$sd=\sqrt{\sum^{\text{​}}\frac{{\left(x_{i}-\bar{x}\right)}^{2}}{\left(n-1\right)}}$$ where $x_{i}$ is the observed value, $\bar{x}$ is the mean of the values, and n is the sample size. The statistical differences between the means were assessed by ANOVA and Tukey’s honestly significant difference (HSD) using a *p*-value of under 0.05 as the threshold of statistical significance. In one case (maximum penetration), the statistical differences between two sample data sets were assessed with Welch Two Sample t-test.

## 3. Results

### 3.1. Curing Behavior of the Adhesives 

Differential Scanning Calorimetry (DSC) was used to analyze the curing behavior of the ALS adhesives crosslinked with pMDI and FOH, with and without tannin (Figure 1, Table 2). For particleboard adhesives, the curing temperature determines if the adhesive fully cures during pressing. The ALS sample had an exothermic curing peak *T*_max1_ of 129 °C. The addition of both crosslinkers—FOH and pMDI—lowered the curing temperature, indicating higher reactivity [47]. ALS-FOH and ALS-FOH + m had *T*_max1_ of 119 °C and 120 °C, respectively. The presence of tannin, however, caused an additional peak to appear at 115 °C for the ALS-FOH + m sample, indicating two separate reactions happening during the heating. The additional reaction can be connected to the reaction of FOH with tannin, which has been reported previously [48]. An additional reaction peak was also detectable for both pMDI crosslinked ALS samples. However, there was a big difference between the samples; the curing of ALS-pMDI happened at the lowest temperature of all the samples (107 °C), while the T_max1_ of ALS-Pmdi + m was the highest of the crosslinked samples at 125 °C. Lignin-based adhesives have varying curing temperatures depending on the crosslinker used. As an example, experimental lignin-formaldehyde resin was found to have curing temperature around 203 °C [49], lignin-phenol-formaldehyde adhesives reported to have curing temperature of 168 °C (75% lignin) and 144 °C (40% lignin) [50], and between 145.7–150.9 °C in another study [47]. Lignin-based epoxy resins had curing temperatures ranging from 90–145 °C, depending on the catalyst and heating rate [51].

The curing heat of ALS was 172 J/g, which is lower than found for lignin extracted from different fiber sources using laboratory organosolv methods [52]. The ALS-FOH had a curing heat of 117 J/g, and the tannin reinforced sample ALS-FOH + m had a curing heat of 126 J/g. The reactivity of furfuryl alcohol-based systems is dependent on the type and amount of catalyst used, as the catalyst needs to reach all the reactive sites in the resin network. In this case, a fixed amount of catalyst (2% of the total solids) was used, and increasing this might have a positive impact on the curing heat based on previous investigation [53]. The ALS-pMDI + m had the highest curing heat (167 J/g), while the ALS-pMDI had the lowest at 123 J/g. This indicated that it took more total energy to cure the ALS-pMDI + m sample than any other sample. This was surprising since pMDI-cured samples typically required lower energy, as in the case of the ALS-pMDI sample. It should be noted that measuring pMDI-containing samples on DSC was challenging, as the reaction with water of the adhesive starts before the temperature was raised. In the case of ALS-pMDI + m, the reaction might have been so fast that the curing had partially happened before the measurement. The curing heats of all of the pMDI and FOH crosslinked ALS samples were lower than that of un-crosslinked, being in the range of 167–117 J/g. They were higher than those reported by Kalami et al. [49] for their experimental lignin-formaldehyde resin (90 J/g) but lower than the commercial phenol-resorcinol-formaldehyde resin used in the same study (171 J/g).

### 3.2. Tensile Shear Strength and Adhesive Penetration

As shown in Figure 2 for the 2-layered veneers, gluing with sole ALS resulted in poor tensile shear strength (0.56 N/mm^2^). The addition of crosslinkers significantly increased the tensile shear strength of the ALS sample to the level of the UmF cured samples (UmF 1.30 N/mm^2^, crosslinked 1.47–1.72 N/mm^2^). There was no significant difference between the crosslinked samples or the UmF bonded sample (*p* > 0.05).

Although the tensile shear strength values were not statistically different among the crosslinked samples, there were differences in the bondline thickness, as seen in Figure 3. The adhesive formulations containing FOH as crosslinker exhibited significantly thicker (*p* < 0.0001) bondlines (ALS-FOH with 52 µm and ALS-FOH + m with 67 µm) than the ones crosslinked with pMDI (ALS-pMDI with 12 µm and ALS-pMDI + m with 26 µm). However, the different bondline morphologies did not seem to play any role in the adhesion, and systems with both crosslinkers were able to bridge the gap between the wood substrates [54]. The two pMDI crosslinked ALS samples did not fill cell lumens beyond the bondline, and thus only the bondline thickness was determined. Adhesive penetration could be observed in adhesive formulations crosslinked with FOH. The MP for ALS-FOH was significantly higher (*p* > 0.05) than for the ALS-FOH + m sample (178 µm and 87 µm, respectively). This could partially be due to the higher viscosity of the ALS-FOH + m sample. Higher viscosities have previously shown to decrease the average penetration depth of adhesives, such as urea-formaldehyde [55]. However, the difference in viscosities was not large enough to fully explain the different penetration behavior. Another explanation can be that tannin increases the crosslinking and contributes to forming larger molecules that prohibit penetration. It should be noted that ALS bonded samples could not be evaluated since wetting of the surfaces with a brush for microtome sectioning resulted in complete delamination. Also, determinations were not performed for the UmF samples as it was aimed to evaluate the behavior of the crosslinked ALS adhesives and not to make any such comparisons.

Microscopic observation of wood-adhesive interphases provided additional information on the penetration of the adhesive formulations (Figure 4a–d). Distinct and continuous bondlines with some voids were observed for the ALS adhesive crosslinked with FOH (ALS-FOH, Figure 4a). Large interphases with continuous and distinct bondlines were observed for the ALS adhesive crosslinked with FOH in the presence of tannin (ALS-FOH + m, Figure 4b). Imperfect and starved bondlines were observed in the case of ALS adhesives crosslinked with pMDI (Figure 4c,d). In addition, the bondlines were lighter in color compared to the FOH crosslinked ALS samples, partially due to the lighter color of pMDI compared to FOH. The lighter color could also be due to the poor crosslinking of pMDI with ALS, leading the lignin to dissolve into the wooden structure when water was added for the sampling. The reaction of isocyanate groups with moisture (in wood or ALS adhesive) results in the production of CO_2_ gas, which causes an inner vapor pressure that drives more adhesive from the bondline towards the wood structure. This phenomenon was reported previously by Bastani and colleagues [40,56]. It has been shown previously that penetration is inversely related to molecular weight, viscosity, and solids content [57]. In this study, the deeper penetration of the adhesive crosslinked with the bio-based crosslinker could be attributed to the lower molecule weight of FOH as compared with pMDI, although the viscosity of the pMDI crosslinked samples was lower. For hardwoods, such as beech, adhesive penetration mainly occurs in longitudinal vessels [56]. In the ALS samples crosslinked with FOH, the penetration mainly occurred through vessels with some filling of fiber lumens. No penetration through rays could be detected. Finding a meaningful relationship between the adhesion penetration results and shear bond performance can be challenging [58]. By looking the microscopy (Figure 3) and tensile strength results (Figure 2), a statistically significant difference in adhesion penetration between the FOH and pMDI crosslinked samples occurred (*p* < 0.0001), but no significant difference (*p* > 0.05) in tensile strength could be detected.

### 3.3. Chemical Characterization of Adhesive Bondline

FTIR spectroscopy analysis was used to monitor changes in the chemical structure of adhesive formulations crosslinked with pMDI upon curing in wood veneers (Figure 5). The stretching vibrations at 2960 and 3410 cm^−1^ were contributed to the aromatic and aliphatic OH groups. ALS alone illustrated two distinct peaks at 1705 and 1720 cm^−1^, which are related to the unconjugated carbonyl stretching vibration [35]. The absorption bands at 1141 and 1180 cm^−1^ were attributed to the aromatic CH bonds in guaiacyl units and stretching of SO groups, respectively. Obvious changes across all regions of the spectra were observed for ALS formulations with the synthetic crosslinker and in combination with tannin, i.e., ALS-pMDI and ALS-pMDI+m. The appearance of peaks at 1712 cm^−1^ and shoulder at 1660 cm^−1^ in the formulations containing pMDI, i.e., ALS-pMDI and ALS-pMDI + m, confirmed the formation of urethane linkages in the bondlines [36,59]. In addition, strong stretching bonds at 1510–1530 cm^−1^ in the ALS-pMDI and ALS-pMDI + m formulations were attributed to urea linkages, which could be due to the reaction of pMDI and water in the adhesive formulations [60]. Formation of urethane and urea bonds proved that all isocyanate groups in pMDI (2270–2250 cm^−1^) were consumed.

### 3.4. Mechanical and Physical Properties of Particleboards

Figure 6 illustrates the internal bond (IB) strength for the particleboards bonded with ALS-based adhesives for synthetic and bio-based crosslinkers, and with or without tannin. It should be mentioned that the panels with sole ALS as binder delaminated after the opening of the press and could not be tested. It was observed that the IB strength in the panels containing pMDI crosslinker was significantly higher than in the panels crosslinked with FOH as well as in controls bonded with UmF (*p* < 0.0001). Crosslinking of ALS with FOH with or without tannin resulted in poor IB results as compared with the control panels, and the differences were statistically significant (*p* < 0.0001). The addition of tannin reduced the IB strength of the panels bonded with ALS-pMDI + m adhesive (*p* < 0.05). However, the IB strength remained unchanged by adding tannin in the panels bonded with ALS-FOH (ALS-FOH + m) (*p* > 0.05). According to EN 312 (2010) [61], the minimum requirement of IB strength of particleboards for indoor applications (P2) in thickness range 6–13 mm is 0.40 N·mm^−2^. The panels bonded with ALS-based adhesives crosslinked with pMDI (ALS-pMDI and ALS-pMDI + m) were above this limit (0.62 N/mm^2^ and 0.49 N/mm^2^, respectively). These values are in line with other pMDI crosslinked lignin adhesives, where 2.5–4% (% on dry wood mass) of pMDI has been used [30]. However, the contribution of lignin into the final strength can be questioned and needs to be further evaluated.

Similar to IB strength, modulus of rupture (MOR) in static bending for the panels was considerably higher for the pMDI crosslinked ALS samples compared to the control UmF (Figure 7, *p* < 0.05). The modulus of elasticity (MOE) of the panels with pMDI was at the same level as the control (*p* > 0.05). However, the panels bonded with ALS-based adhesive crosslinked with FOH showed significantly lower MOR and MOE as compared with the formulation containing pMDI (*p* < 0.0001) and with the UmF control (*p* < 0.0001). The addition of tannin did not change the MOR and MOE in the particleboards bonded with adhesives having FOH or pMDI as crosslinkers. The highest bending properties were obtained from panels bonded with ALS-pMDI (MOR: 11.1 N/mm^2^, MOE: 2056 N/mm^2^) and ALS-pMDI + m (MOR: 10.9 N/mm^2^, MOE: 1856 N/mm^2^). According to EN 312 [61], the respective minimum requirements for MOR and MOE of particleboards in thickness range 6 to 13 mm for indoor applications are 11 N/mm^2^ and 1800 N/mm^2^. No difference in mechanical properties could be detected with the addition of 10% of tannin to the lignin-FOH copolymer, and that was also true for the tensile shear strength of the 2-layered beech samples. This is in contradiction to the pre-mentioned study by Luckender et al. (2016) [62], where for FOH copolymers, the best results could be obtained by using spent liquor lignin and tannin together with FOH, in oppose to using only lignin or only tannin with FOH. Best results were obtained with spent liquor lignin: tannin ratios of 50:10 and 40:20 (IB 0.55 N/mm^2^ and 0.53 N/mm^2^, respectively, with adhesive addition amount of 15%).

All mechanical results of particleboards were poor for the FOH crosslinked ALS samples. However, in the case of 2-layered beech samples, the tensile shear strength was at the same level for reference and all the crosslinked ALS samples. One explanation could be the difference in the FOH pre-polymerization between the veneer and particleboard samples. The increase of viscosity of furfuryl copolymers happens very rapidly and is highly affected by small differences in the amount of hardener, temperature, and time [62]. In order to keep the viscosity at a constant level suitable for the spray nozzle to reach even distribution in the chip blender, FOH was only reacted for 30 min in the adhesive formulation before resination. This was shorter than the reaction time of 45 min used for gluing the 2-layered beech samples. Thus, FOH had less time for pre-polymerization before board pressing compared to the veneer pressing. In a study by Dao et al. [63], the effect of polymerization time on the final furfuryl alcohol strength was shown. Different polymerization times of polyfurfuryl alcohol were tested for wood powder composites, and increasing the polymerization time from 30 min to 110 min resulted in an increase of the dry tensile shear strength from 2450 psi (16.9 N/mm^2^) to 3730 psi (25.7 N/mm^2^), and the wet tensile shear strength from 1270 psi (8.7 N/mm^2^) to 1960 psi (13.5 N/mm^2^). Another explanation for the low strength of the FOH crosslinked ALS samples could be the low addition amount for gluing the particleboards. A previous study reported that the bonding quality of the panels is directly related to the amount of FOH and catalyst in the adhesive formulation. In this study, a total amount of 40% of FOH was used in making the particleboards, and the total adhesive amounts were 10% and 15% [62]. In the current study for particleboard manufacturing, it was aimed to keep the amounts of pMDI and FOH crosslinker the same (25% of total adhesive) at the industrially relevant level of total added adhesive (8 wt % to dry particles). Therefore, both the total adhesive amount and the FOH share in it were lower than in the previously mentioned study.

The thickness swelling (TS) of the particleboards bonded with UmF and different ALS-based adhesive systems are presented in Figure 8. The reference boards showed TS of 40% and 52% measured after 2 h and 24 h, respectively. The panels bonded with ALS-FOH and ALS-FOH + m were disintegrated after 2 h and thus could not be measured. There was no significant difference between the ALS-FOH and ALS-FOH + m samples (*p* > 0.05). The sensitivity of FOH towards the water was reported previously by Luckender et al. [62]. The samples with the synthetic crosslinker pMDI had significantly lower TS compared to the reference UmF (*p* < 0.0001 for 2 h swelling and *p* < 0.05 for 24 h swelling). The isocyanate groups in pMDI are highly unsaturated and can react with a number of active hydroxyl groups in wood chips’ surface and ALS structure, as well as with the moisture contained in the chips and the glue system. That will result in the formation of polyurethane and polyurea linkages, respectively [64]. The presence of excess primary amino-groups at elevated temperature will also lead to allophanate and/or biuret bonds [65]. The mentioned reactions provide durable linkages for adhesive systems containing pMDI crosslinker under moist conditions. One possible way to increase the water-resistance of the lignin-FOH adhesives is the addition of glyoxal to the lignin-FOH copolymer. Zhang et al. [27] found that furfuryl alcohol does react with glyoxal, and the addition of glyoxal has a positive effect on the dry strength and water resistance of lignin-FOH copolymers. The addition of epoxy further increased the properties to a level where lignin-FOH glyoxal adhesive with 9% of epoxy had IB of 0.41 MPa and IB after 2 h in boiling water of 0.21 MPa.

### 3.5. Formaldehyde Emissions

In this work, the previously evaluated method based on a dynamic microchamber (DMC), according to ASTM D 6007 [44], after 1-day conditioning [45] was used to determine the extreme formaldehyde emission values of the particleboard samples (Figure 9).

The panels bonded with commercial adhesive (UmF) had a formaldehyde emission value of 0.073 ppm. The emissions were lower for all ALS-based adhesive systems. The lowest emissions were obtained for the boards bonded with ALS-pMDI (0.027 ppm). The panels bonded with ALS-FOH exhibited a considerably higher formaldehyde emission value of 0.059. This can be explained by the polycondensation mechanism of FOH, where the reaction between two methylol groups in the FOH monomer leads to the formation of ether bridge that can transform into methylene bridge by releasing formaldehyde molecule [23,48] (Figure 10). The formaldehyde emissions of the particleboards crosslinked with pMDI remained unchanged by tannin addition. However, emissions decreased to a minor extent in the boards crosslinked with FOH. The formaldehyde scavenger effect of tannin was reported previously [66].

## 4. Conclusions

In this work, two crosslinkers for ALS, with and without mimosa tannin, were evaluated—furfuryl alcohol (FOH) and polymeric 4,4′-diphenylmethane diisocyanate (pMDI). Curing of the adhesive systems was performed using DSC, and properties of the crosslinked ALS adhesives were tested on 2-layered beech veneers and 3-layer particleboards. The conclusions can be summarized as follows:All samples with crosslinkers had lower curing temperature and curing heat than the un-crosslinked ALS sample. The ALS-pMDI sample cured at the lowest temperature (107 °C) and had the lowest curing heat (123 J/g).For those formulations crosslinked with pMDI, FTIR results pointed out for chemical reaction of ALS with pMDI in the cured adhesives, i.e., the formation of urethane linkages.The FOH crosslinked ALS samples had thick dark bondlines, while the pMDI cured ALS samples had thin light bondlines, possibly due to the higher molecular weight of pMDI and vapor pressure created by the isocyanate crosslinking.The penetration of the FOH crosslinked ALS occurred mainly through vessels with some filling of the fiber lumens. The FOH crosslinked ALS without tannin had deeper penetration than the one with tannin, suggesting that tannin prohibits penetration, allowing more glue to remain near the bondline.The mechanical properties of the particleboards produced using FOH crosslinked ALS, with and without tannin (IB: 0.17 N/mm^2^ and 0.18 N/mm^2^, respectively), were inferior to those produced using pMDI crosslinked ALS, with and without tannin (IB: 0.62 N/mm^2^ and 0.49 N/mm^2^, respectively). A similar trend could be seen in thickness swelling where the FOH crosslinked ALS samples disintegrated after 2 h. The contribution of ALS to the final strength of the ALS pMDI samples needs to be further evaluated.Although particleboard properties were worse for the FOH crosslinked samples, the tensile shear strengths of both FOH and pMDI crosslinked ALS were at the same level as the UmF reference for the 2-layered veneer samples. The poor performance of FOH crosslinked ALS samples in particleboards could be due to the shorter pre-polymerization time and low FOH and total glue amount.The addition of mimosa tannin (10% to ALS amount) had no effect on any of the mechanical properties for both particleboard and 2-layered veneer samples. However, it lowered the emissions of FOH crosslinked ALS samples from 0.059 ppm of ALS FOH to 0.050 ppm of ALS FOH + m.The formaldehyde emissions of pMDI crosslinked particleboards were at the level of natural wood (0.027 ppm and 0.028 ppm). Interestingly, the FOH crosslinked boards emitted beyond the level of natural wood (0.059 ppm and 0.050 ppm). This can be due to one of the polycondensation mechanisms of FOH, where two methylol groups of two FOH molecules react, leading to ether bridge that can transform into methylene bridge by the release of formaldehyde.

No particleboards could be produced using the ALS without crosslinker, highlighting the need for a crosslinker if the lignosulfonate is used without modification. More work in finding suitable crosslinkers for bio-based materials is needed, but until then, pMDI is a promising crosslinker that works for most bio-based adhesives. This study demonstrated the potential to combine ALS and pMDI in particleboard manufacturing; however, the chemical interaction between the polymers needs to be further elucidated for optimum usage. Thus, further work in lowering the pMDI amount, optimizing pressing parameters, and modifying the lignosulfonate formula should be done as the next step to confirm that the ALS is truly contributing to the final adhesion strength.

## Figures and Tables

**Figure 1 polymers-11-01633-f001:**
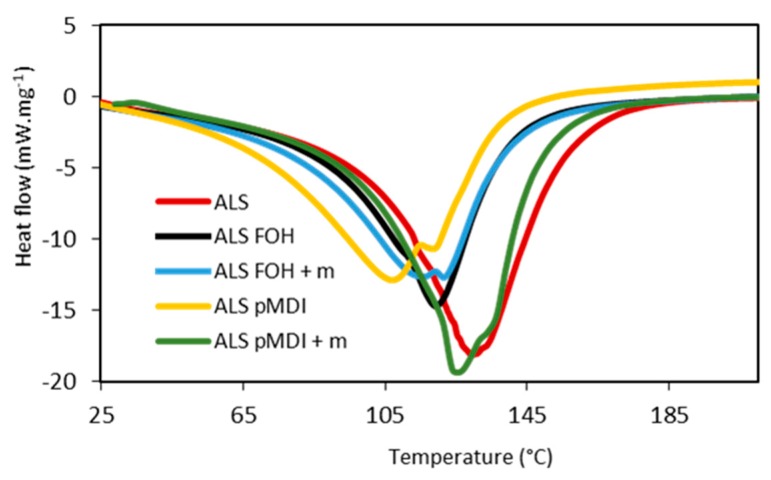
DSC heat of reaction thermograms of adhesive systems based on ammonium lignosulfonate (ALS), furfuryl alcohol (FOH), and polymeric 4,4′-diphenylmethane diisocyanate (pMDI) as crosslinkers, and with or without mimosa tannin (m).

**Figure 2 polymers-11-01633-f002:**
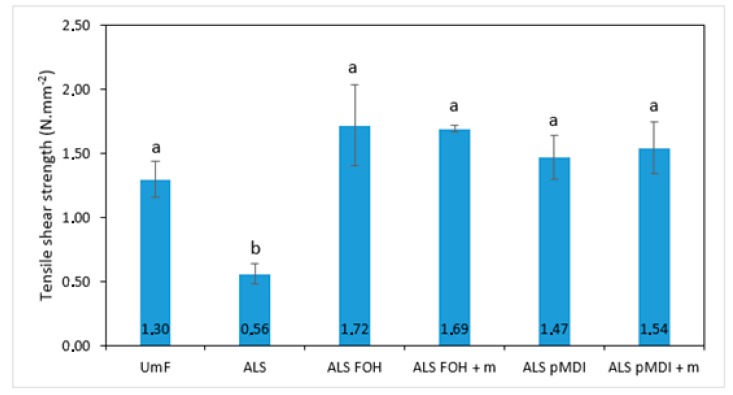
Tensile shear strength of 2-layered beech samples bonded with urea-melamine-formaldehyde (UmF), ammonium lignosulfonate (ALS), ALS crosslinked with furfuryl alcohol (ALS FOH), ALS FOH with mimosa tannin (ALS FOH + m), ALS crosslinked with pMDI (ALS pMDI) and ALS pMDI with mimosa tannin (ALS pMDI + m). Each column is the average of 10 determinations, and error bars represent standard deviations. Values labeled with different letters (a and b) are statistically different at an error probability of α = 0.05 (ANOVA and Tukey’s HSD tests). UmF, melamine reinforced urea-formaldehyde; HSD, honestly significant difference.

**Figure 3 polymers-11-01633-f003:**
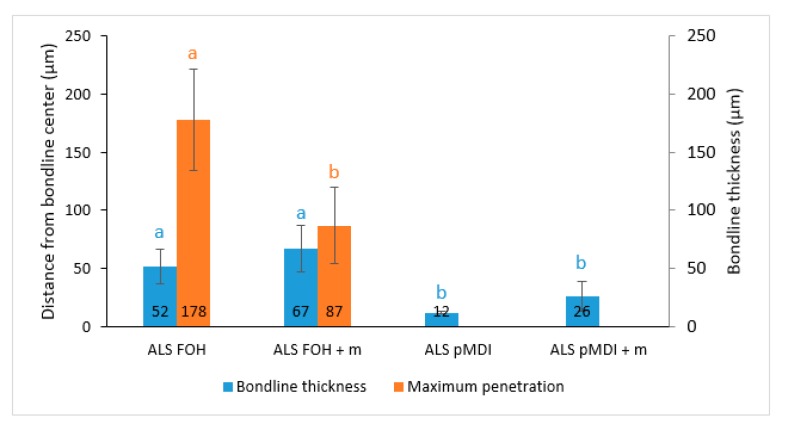
Bondline thickness (blue bars) and maximum penetration (orange bars) in 2-layered beech samples bonded with ammonium lignosulfonate (ALS) crosslinked with furfuryl alcohol (ALS FOH), ALS FOH with mimosa tannin (ALS FOH + m), ALS crosslinked with pMDI (ALS pMDI) and ALS pMDI with mimosa tannin (ALS pMDI + m). Each bondline thickness and maximum penetration column is the average of 15 and 30 determinations, respectively. Error bars represent standard deviations. Values labeled with different letters (a and b) are statistically different at an error probability of α = 0.05 (bondline thickness ANOVA and Tukey’s HSD tests, maximum penetration Welch Two Sample t-test).

**Figure 4 polymers-11-01633-f004:**
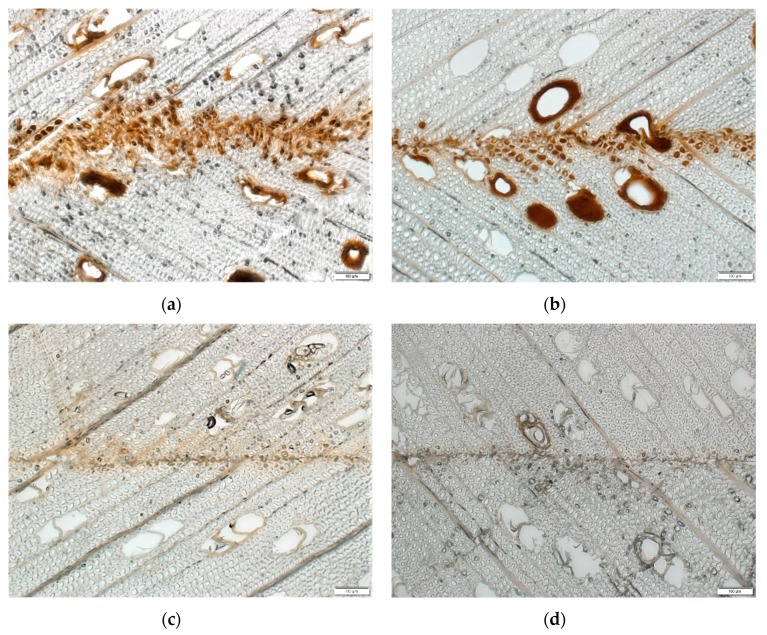
Transverse visible-light view of horizontal bondlines in 2-layered veneer samples: (**a**) ALS-FOH, (**b**) ALS-FOH +m, (**c**) ALS-pMDI, and (**d**) ALS-pMDI + m.

**Figure 5 polymers-11-01633-f005:**
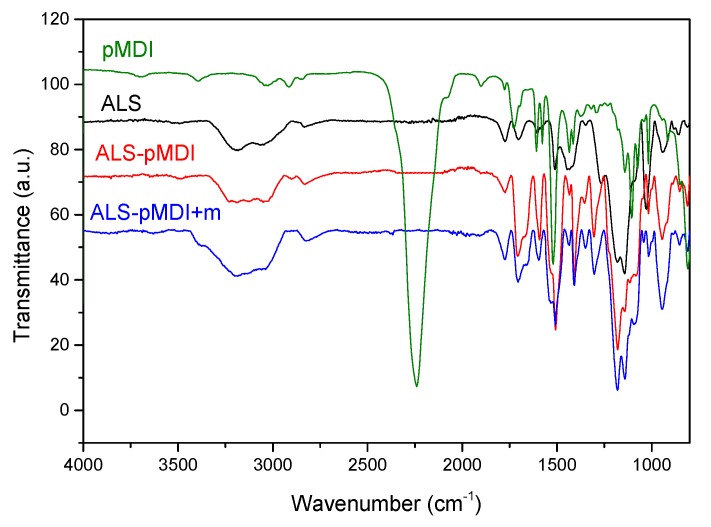
Fourier transform infrared spectroscopy (FTIR) spectra of pure pMDI and cured adhesive formulations with ALS, ALS-pMDI, and ALS-pMDI + m.

**Figure 6 polymers-11-01633-f006:**
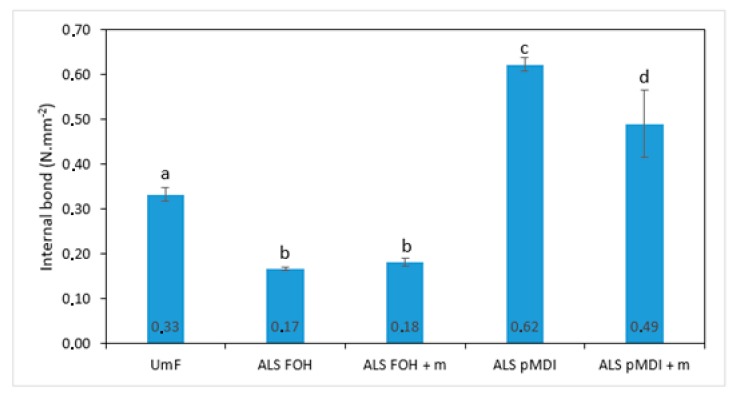
Internal bond strength of particleboard panels produced with urea-melamine-formaldehyde (UmF), ammonium lignosulfonate (ALS) crosslinked with furfuryl alcohol (ALS FOH), ALS FOH with mimosa tannin (ALS FOH + m), ALS crosslinked with pMDI (ALS pMDI) and ALS pMDI with mimosa tannin (ALS pMDI + m), and with or without mimosa tannin (m). Error bars represent standard deviations. Values labeled with different letters (a, b, c and d) are statistically different at an error probability of α = 0.05 (Tukey’s HSD test).

**Figure 7 polymers-11-01633-f007:**
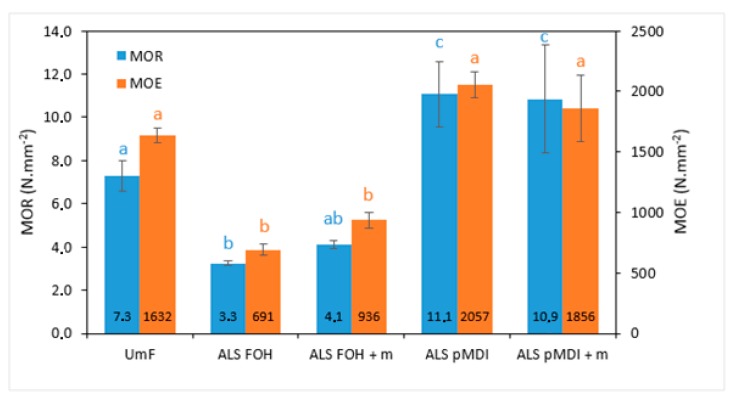
MOR (modulus of rupture, blue bars) and MOE (modulus of elasticity, orange bars) of particleboards produced with urea-melamine-formaldehyde (UmF), ammonium lignosulfonate (ALS) crosslinked with furfuryl alcohol (ALS FOH), ALS FOH with mimosa tannin (ALS FOH + m), ALS crosslinked with pMDI (ALS pMDI) and ALS pMDI with mimosa tannin (ALS pMDI + m). Error bars represent standard deviations. Values labeled with different letters (a, b, and c) are statistically different at an error probability of α = 0.05 (ANOVA and Tukey’s HSD tests). Value marked ab is not statistically different from a or b. Different colored letters refer to the matching mechanical property.

**Figure 8 polymers-11-01633-f008:**
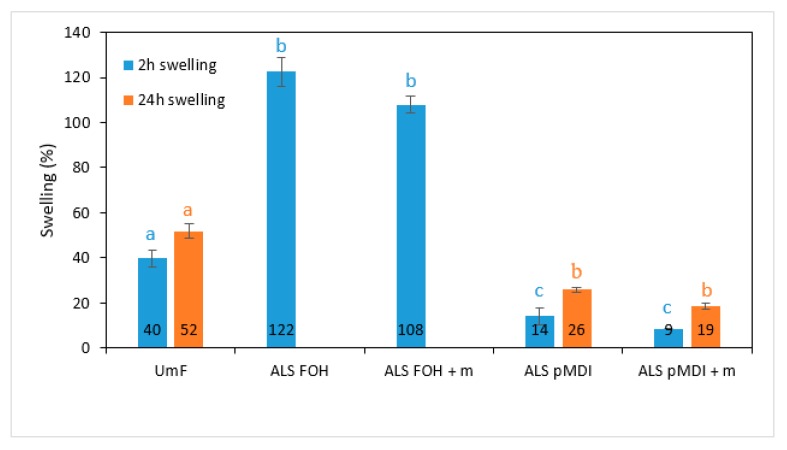
Thickness swelling of particleboards produced with urea-melamine-formaldehyde (UmF), ammonium lignosulfonate (ALS) crosslinked with furfuryl alcohol (ALS FOH), ALS FOH with mimosa tannin (ALS FOH + m), ALS crosslinked with pMDI (ALS pMDI) and ALS pMDI with mimosa tannin (ALS pMDI + m) after two hours (blue bars) and after 24 hours (orange bars). Error bars represent standard deviations. Values labeled with different letters (a, b and c) are statistically different at an error probability of α = 0.05 (ANOVA and Tukey’s HSD tests). Different colored letters refer to swelling property.

**Figure 9 polymers-11-01633-f009:**
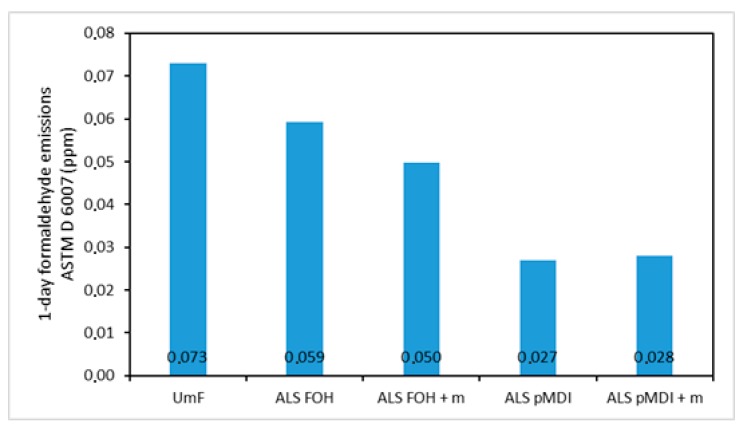
Formaldehyde emissions of particleboards produced with urea-melamine-formaldehyde (UmF), ammonium lignosulfonate (ALS) crosslinked with furfuryl alcohol (ALS FOH), ALS FOH with mimosa tannin (ALS FOH + m), ALS crosslinked with pMDI (ALS pMDI) and ALS pMDI with mimosa tannin (ALS pMDI + m).

**Figure 10 polymers-11-01633-f010:**
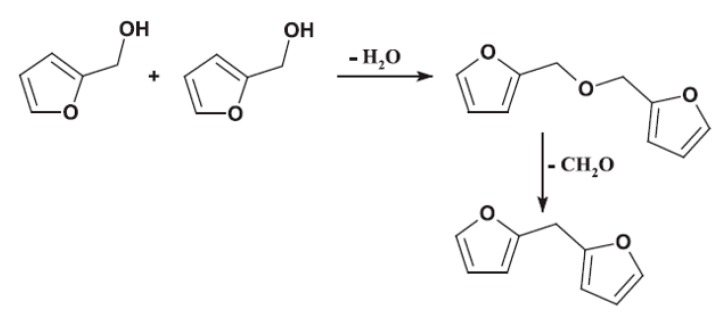
One of furfuryl polycondensation pathways: the reaction between the methylol groups of two furfuryl alcohol molecules, leading to dimethyl ether bridge that can be transformed into a methylene bridge by the release of formaldehyde [48].

**Table 1 polymers-11-01633-t001:** Adhesive compositions for testing of 2-layered veneer samples and particleboards based on ammonium lignosulfonate (ALS), furfuryl alcohol (FOH), and polymeric 4,4′-diphenylmethane diisocyanate (pMDI) as crosslinkers, and with or without mimosa tannin (m).

Adhesive ID	Base ^1^: Lignin to Tannin Ratio	Crosslinker Amount in Relation to Base ^1^ (%)	Application Amount for 2-Layered Veneer (g solids/m^2^)	Application Amount for Particleboard (wt % to dry particles)	Final Viscosity for Veneer/Particleboard (mPa·s)
UmF ^2^			100	12	370/370
ALS	10:0	-	100	12	80/90
ALS-FOH	10:0	25	100	8	2100/210
ALS-FOH + m	9:1	25	100	8	2350/190
ALS-pMDI	10:0	25	100	8	120/110
ALS-pMDI + m	9:1	25	100	8	110/100

^1^ Base refers to lignin or the lignin-tannin mixture, ^2^ Melamine-urea formaldehyde with 3% (*w/w* of dry resin) of ammonium sulfite as a hardener.

**Table 2 polymers-11-01633-t002:** Details on the DSC thermograms of adhesive systems based on ammonium lignosulfonate (ALS), furfuryl alcohol (FOH), and pMDI as crosslinkers, and with or without mimosa tannin (m).

Adhesive identification	Onset (°C)	*T*_max1_ (°C)	*T*_max2_ (°C)	Curing Heat (J/g)
ALS	104	129	-	172
ALS-FOH	98	119	-	117
ALS-FOH + m	84	120	115	126
ALS-pMDI	75	107	118	123
ALS-pMDI + m	113	125	135	167

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
