# Peer review of "Ammonium Lignosulfonate Adhesives for Particleboards with pMDI and Furfuryl Alcohol as Crosslinkers"

_polymers, 2019, doi:10.3390/polym11101633_

Round 1

Reviewer 1 Report

The paper is good, the idea is interesting, it's all well done, but some things need to be explained and added.

175 - 190 too extensive explanation of the principles of shear bond test, for experienced readers it is not necessary – it should to be shorten

178 – what species and why moisture content of veneers 12+/-1 % ?- it is too high

178 – why 2-layers samples only? For shear bond test OK, but what about other mechanical properties of thicker plywood?

215 – what species? – it should be explained

217 – why no wax if “standard industry conditions” for particleboard production where used? – it should be explained

219 - why 65 % core share if “standard industry conditions” for particleboard production where used? – it should be explained

219 - why 12 mm if “standard industry conditions” for particleboard production where used? – it should be explained

220 – why target density 620 kg/m3 if “standard industry conditions” for particleboard production where used? – it should be explained

233 – why the method ASTM D6007-14 was used? – it should be explained

30-123 and 231-244 - it is also good to know about the possibilities of tightening of formaldehyde emissions by the modifying adhesive mixtures of UF adhesives, e.g. by adding active powdered bark, the results are very promising, we recommend to mention it, for example:

https://www.mdpi.com/1996-1944/12/8/1298

https://link.springer.com/article/10.1007/s00107-016-1096-0

Reviewer 2 Report

Its an interesting article to make a less toxic adhesive material. The author used MDI as a crosslinker, which ultimately make urethane by using polyol. However, its need to improve before the publication counting the following points:

The same adhesive material can be used for other application. Thus its need to analyze and explain the bonded urethane group by FT-IR analysis. Its also need to confirm the absence of isocyanate group from MDI. Any unreacted MDI can make allophanate group at the mentioned condition. Allophanate may have detrimental effect on adhesive strength. Optimized MDI content is also very important. Excess crosslinking make brittleness and decrease the adhesive strength. The author can use different MDI content to have the best ratio.  The pot life is very important for adhesive material. Please mention the pot life when MDI was mixed with ALS-water solution.   

Reviewer 3 Report

Dear Editor,

This manuscript describes the use of ammonium lignosulfonate as an adhesive for particle boards using pMDI and furfuryl alcohol as crosslinkers. The aim is clearly described, the experiments are well designed, and the results are interesting and well explained. I recommend accepting the manuscript upon minor corrections:

The introduction is nicely thorough but very long. Many parts of it should be removed/shortened including figure 1 as the reader is expected to know the structural units of lignin. Please define the term UmF in the abstract before using it. Line 36: “….lowering the European E1 emission level of 0.01 to 0.05 ppm”. Please check the numbers again! Figure 9: Y-axis: correct “missions” to “emissions”. Line 10: “has boosted” should be “have boosted”. Table 2: are these results reproducible? Please add standard deviation. Please also consider significant figures when reporting the temperature values. For instance, you write 104.0 C as the onset tempearture. You should report it as 104 C.

Round 2

Reviewer 2 Report

Though the author mentioned the future research on my previous comments, however, at least it needs to confirm that there was no unreacted MDI by FT-IR analysis. 
